# The Active Role of the Internet and Social Media Use in Nonpharmaceutical and Pharmaceutical Preventive Measures against COVID-19

**DOI:** 10.3390/healthcare10010113

**Published:** 2022-01-06

**Authors:** Tian Xie, Meihui Tang, Robert Jiqi Zhang, James H. Liu

**Affiliations:** 1Department of Psychology, Philosophy School, Wuhan University, Wuhan 430072, China; thanksky520@whu.edu.cn (T.X.); 2018301131030@whu.edu.cn (M.T.); 2School of Psychology, Nanjing Normal University, Nanjing 210024, China; r.j.zhang@nnu.edu.cn; 3School of Psychology, Massey University, Auckland 0745, New Zealand

**Keywords:** internet, social media, COVID-19, misinformation, preventive measures, perceived threat

## Abstract

During the COVID-19 pandemic, does more internet and social media use lead to taking more- or less-effective preventive measures against the disease? A two-wave longitudinal survey with the general population in mainland China in mid-2020 found that during the COVID-19 pandemic, internet and social media use intensity promoted the adoption of nonpharmaceutical and pharmaceutical antipandemic measures. The first wave of data (*n* = 1014) showed that the more intensively people used the internet/social media, the more they perceived the threat of the pandemic, and took more nonpharmaceutical preventive measures (e.g., wearing masks, maintaining social distance, and washing hands) as a result. The second wave (*n* = 220) showed firstly the predicted relationship between internet/social media use intensity and the perceived threat of the pandemic and the adoption of nonpharmaceutical preventive measures by cross-lagged analysis; secondly, the predictive effect of internet/social media use on the adoption of pharmacological measures (i.e., willingness to vaccinate against COVID-19) and the mediating role of perceived pandemic threat were verified. The article concludes with a discussion of the role of the internet and social media use in the fight against COVID-19 in specific macrosocial contexts.

## 1. Introduction

As a global public health emergency, the COVID-19 pandemic had caused more than 4.8 million deaths with more than 239 million confirmed cases globally as of 16 October 2021 [1]. When the outbreak first began in early December 2019 in Wuhan [2,3], people lacked a sufficiently accurate understanding of the pandemic—where did the virus come from, how did it spread, and how could it be prevented?

However, as the pandemic has developed, research on it has also intensified, and scientists have gradually come to a better understanding of it. For the public, at this point, the problem they are facing is how to form correct judgments and take the right actions against the pandemic based on credible information. Online social media is the most-searched venue for information gathering and sharing [4]. As technology has progressed, the boundaries between social media and the internet have become increasingly blurred, so the two terms were not distinguished in the current study, as in previous research [5]. An interesting question is: if people use the internet and social media to obtain pandemic-related information, will they have a misperception of the pandemic and act against their own best interests and against the best interests of society? Or, does internet and social media use allow people to make a correct judgment about the severity of the pandemic and take effective preventive measures against it?

It is believed that every major disease outbreak is accompanied by a high volume of misinformation, which is amplified by the internet and social media. This phenomenon has been vividly named as “infodemic” [6], which needs to be fought against or managed [7]. It was found that the infodemic curve fluctuated with the epidemic curve [8]. Given the large amount of false information online, people can easily become misinformed [9], especially about COVID-19 [10]. Previous studies on online information concerned mainly false information, termed as misinformation [11], disinformation [12], and rumors (circulating stories or reports of uncertain or doubtful truth) or “fake news” (i.e., news emanating from websites that falsely claim to be news organizations while publishing deliberately false stories for garnering advertising revenue) [13]. Since the current study was not concerned with the intent of the creation and sharing of false information, misinformation was used as a conceptual umbrella to include all kinds of false information related to COVID-19, as in previous studies [14].

Online misinformation may cause people to form incorrect judgments about the pandemic and take inappropriate actions against it. For example, the appropriate way of protecting oneself from the pandemic is to rely on nonpharmaceutical measures (e.g., washing hands regularly, social distancing, mask usage, and adopting contact tracing systems [15]) in addition to pharmaceutical preventive measures (e.g., vaccination). However, there is much antimask messaging posted on Twitter, even though the Centers for Disease Control and Prevention (CDC) in the USA and other public health experts have repeatedly indicated, on multiple occasions, that wearing masks could save a significant number of lives [16]. Similarly, although the adoption of vaccination has been recommended by public health institutions, misinformation and disbelief in vaccination (including conspiracy theories) are important barriers that are preventing control of the pandemic [17]. Therefore, it may be natural to infer that heavy internet and social media usage may lead to inaccurate perception of the severity of the pandemic, and incorrect measures against it as a result.

However, empirical studies also suggest that useful information is more likely to be passed on than misinformation, as useful stories [18] and marketing messages [19] are more likely to be shared. Consistent with this reasoning, psychologists have long theorized that one of the reasons why people share misinformation (such as rumors, folktales, and urban legends) is “because they seem to convey true, worthwhile and relevant information” [20,21], and when people “desire to benefit and help others”, they may share negative evaluative information (such as gossip) [22]. One recent study used unique behavioral data on Facebook activity linked to individual-level survey data, and found that sharing misinformation was relatively rare during the 2016 U.S. election campaign [23]. Another recent study surveyed more than 1000 Irish residents, and found that most respondents had a good level of knowledge and practices of pandemic prevention [24]. The availability and high accessibility of credible information on the internet and in social media may be due to the fact-checking endeavors made by online platforms. In mainland China, for example, special antirumor accounts on Weibo (China’s equivalent to Twitter) are dedicated to debunking rumor content, and the platform even removes messages carrying misinformation [25,26]. Considering the availability of credible information sources after the early outbreak of the COVID-19 in general [27], and the fact-checking strategy in mainland China in particular, we expected that Chinese people could obtain useful information about how to prevent the pandemic on the internet and in social media.

How can people benefit from the obtained useful information online? Many studies have confirmed a significant relationship between receiving information via social media and perceived pandemic threat. It was found that repetitive exposure to information about a crisis promoted individuals’ perception of it as a threat [28]. Another study on the Middle East respiratory syndrome coronavirus (MERS-CoV) suggested that receiving relevant and specific information about this illness via social media was positively related to its perceived threat as well [29]. A recent study also showed the cognitive factors of the risk perception regarding COVID-19 played an important role in promoting people’s reported compliance to official antipandemic behavioral provisions [30]. Putting this research together, we hypothesized that greater internet and social media use would be related to more willingness to take preventative measures against the disease to the extent that internet use increases pandemic threat perception. In other words, we expected that pandemic threat perception should mediate the relationship between internet/social media use and willingness to take preventative measures.

To test the above hypotheses, a two-wave longitudinal study was designed using the general population in mainland China. The first-wave survey was taken in July 2020, and was used to examine the mediating role of the perception of pandemic threat between the internet and social media use intensity and the adoption of nonpharmaceutical preventive measures (e.g., wearing masks, maintaining social distance, and washing hands regularly). The second-wave survey, using the same participants, was taken three months later, in October 2020. Through a cross-lagged design, we wanted firstly to provide more evidence for our hypothesized predicted relationship of internet/social media intensity leading to the perception of pandemic threat and adoption of nonpharmaceutical preventive measures, rather than the other way around. Secondly, we intended to verify the results pattern of Wave 1 conceptually by measuring the pharmaceutical preventive measures (i.e., willingness to be vaccinated against COVID-19) in Wave 2.

## 2. Materials and Methods

### 2.1. Participants and Procedures

We designed the survey in Qualtrics, an online survey platform, and contracted a commercial survey company (Toluna) to send out survey invitations to people from their commercially available panel via electronic means, with quotas set for gender, age, and socioeconomic status, to try to match national census demographics. After completing the survey in July 2020 (Wave 1), participants were invited to complete a follow-up survey after 3 months (Wave 2). In the first wave of data collection (Wave 1), we recruited 1017 participants (*M_age_* = 37.25, *SD* = 10.32; age range = 18–83; 48.3% male, 51.5% female, with 2 unreported) from the large scale via Toluna. Among them, 220 (*M_age_* = 36.65, *SD* = 8.69; age range = 21–69; 47.3% male, 52.3% female, with 1 unreported) participated in Wave 2, with a 78.4% attrition rate from Wave 1 to Wave 2.

Internet/social media intensity, the perceived threat of COVID-19, the intention of taking nonpharmaceutical preventive measures, online time, and demographics (including age, gender, education, and subjective socioeconomic status), were collected in Wave 1. Internet/social media intensity, the perceived threat of COVID-19, willingness to take nonpharmaceutical and pharmaceutical preventive measures, online time, and subjective socioeconomic status were collected in Wave 2.

### 2.2. Measures

Internet/social media intensity was measured in Wave 1 and Wave 2 by five items, which were based on our previous studies [31], and took into account the specifics of internet use in China [32]: “How frequently do you use the internet for (1) connecting socially to others; (2) getting political news and information; (3) sharing and information with others; (4) making commercial transactions; (5) entertainment”. Respondents’ answers were coded as: 1 = never, 2 = rarely, 3 = somewhat rarely, 4 = occasionally, 5 = somewhat frequently, 6 = frequently, and 7 = always. The average score was used (*α*_1_ = 0.79, *α*_2_ = 0.80).

The perceived threat of COVID-19 was measured in both waves by a single item: “COVID-19 is a serious threat to public safety around the world.” Respondents’ answers were coded as: 1 = disagree completely, 2 = disagree, 3 = disagree a little, 4 = neutral, 5 = agree a little, 6 = agree, and 7 = agree completely.

Self-reported taking of nonpharmaceutical preventive measures was measured in Wave 1 by one item: “I have taken preventive measures as a protection against COVID-19 (e.g., wearing a mask, staying at home as much as possible, washing hands regularly).” Respondents’ answers were coded as: 1 = disagree completely, 2 = disagree, 3 = disagree a little, 4 = neutral, 5 = agree a little, 6 = agree, and 7 = agree completely.

Willingness to take pharmaceutical preventive measures was measured in Wave 2 by one item: “If a COVID-19 vaccine is discovered, I would be willing to be vaccinated.” Respondents’ answers were coded as: 1 = disagree completely, 2 = disagree, 3 = disagree a little, 4 = neutral, 5 = agree a little, 6 = agree, and 7 = agree completely.

Online time was measured in both waves by two items: “How many hours a day would you say you are online (i.e., connected to the internet)?” The respondents’ answers were coded as: 1 = almost all waking hours, and 2 = numbers of hours per day. After completing this question, participants who responded “2” reported the specific amount of time they were online per day.

Demographics were measured in Wave 1, including participants’ age (in years), gender (1 male, 2 female), education (1 elementary school, 2 middle school, 3 high school, 4 some college, 5 bachelor’s degree at university, 6 graduate degree or higher). Participants’ self-reported subjective socioeconomic status was measured in both Wave 1 and Wave 2 using a single item: “On a scale of 1 to 10, with 10 being people who are the most well off in society, and 1 being the people who are least well off, where would you describe your position?”.

### 2.3. Data Analysis

SPSS 26.0 [33] and Mplus 8.6 [34] were used to analyze the data in this study. First, descriptive statistics and correlations between variables were calculated. Next, analysis of the appropriateness of assumptions of normality, linearity, independence, and homoscedasticity were conducted, and assumptions for parametric analyses were supported. Then, we conducted a mediation analysis through the PROCESS macro (version 4.0) on SPSS 26.0 to ascertain the mediation effect of the perceived threat of COVID-19 between internet/social media intensity and the intention of taking nonpharmaceutical preventive measures (model 4; 5000 bootstrapped samples). Age, gender, education, and subjective socioeconomic status were included as covariates.

Cross-lagged analysis was employed to examine the predicted relationship between internet/social media intensity and the perceived threat of COVID-19, and the adoption of nonpharmaceutical preventive measures. All cross-lagged analyses were conducted in Mplus 8.6. Age, gender, education, and subjective socioeconomic status were included as covariates. Missing data were not imputed; rather, available data from all 217 participants were used in analyses. To estimate the models’ goodness-of-fit, we also report the chi-square, comparative fit index (CFI), root-mean-square error of approximation (RMSEA), and standardized root-mean-square residual (SRMR). A CFI close to or above 0.95 and an RMSEA close to or lower 0.06, combined with an SRMR of 0.08 or lower, indicated a reasonably good fit [35,36].

Finally, to test if results in the nonpharmaceutical condition could be replicated in the pharmaceutical condition (willingness to be vaccinated), we further conducted a mediation analysis with the PROCESS macro (version 4.0) to ascertain the mediation effect of the perceived threat of COVID-19 between internet/social media intensity and willingness to take pharmaceutical preventive measures (model 4; 5000 bootstrapped samples) [37]. Age, gender, education, and subjective socioeconomic status were included as covariates.

### 2.4. Sample Attrition

We conducted a sample attrition analysis following Fitzgerald et al. [38], which is widely used [39,40]. Those who participated during only Wave 1 (*n* = 797) did not differ from those who participated in both waves (*n* = 220) in mean age (*F* (1, 1012) = 201.53, *p* = 0.17, partial *η*^2^ < 0.01), gender (*χ*^2^ (1, *n* = 1015) = 0.03, *p* = 0.87), education (*χ*^2^ (1, *n* = 1014) = 6.272, *p* = 0.28), or subjective socioeconomic status in Wave 1 (*F* (1, 1013) = 1.473, *p* = 0.23, partial *η*^2^ < 0.01). Additionally, people sampled during Wave 1 did not differ from those who were sampled during both waves on Wave 1 measures of internet/social media use (*F* (1, 1014) = 0.84, *p* = 0.36, partial *η*^2^ < 0.01), the perceived threat of COVID-19 (*F* (1, 1015) = 1.33, *p* = 0.20, partial *η*^2^ < 0.01). Although sample attrition was substantial, these results indicate little evidence of attrition effects (see also Table 1 and Table 2).

## 3. Results

The results are presented in four subsections. The first subsection reports descriptive statistics and correlations between variables. The second subsection presents mediating effects of the perceived threat of COVID-19 on relationships between internet/social media intensity and the intention of taking nonpharmaceutical preventive measures. The third subsection shows the results of the cross-lagged analyses. The final subsection presents the mediating effects of the perceived threat of COVID-19 on relationships between internet/social media intensity and willingness to take pharmaceutical preventive measures.

### 3.1. Descriptive Statistics

The correlations, means, and standard deviations among variables are presented in Table 1 and Table 2. Table 1 presents the associations between internet/social media intensity and the perceived threat of COVID-19, and the intention of the nonpreventive measures in Wave 1. Table 2 presents the associations between internet/social media intensity and willingness to take pharmaceutical preventive measures (i.e., the COVID-19 vaccine).

In Wave 1, 48.1% of participants reported they were online almost all waking hours, with 3 unreported; and 51.6% reported their everyday online hours (*M* = 5.79, *SD* = 2.80, *Mdn* = 5.00, with 79 unreported). In Wave 2, 53.2% reported they were online almost all waking hours, and 46.8% reported their everyday online hours (*M* = 6.33, *SD* = 3.20, *Mdn* = 5.50, with 11 unreported).

### 3.2. Mediation Analysis for Nonpharmaceutical Measures by Wave 1 Data

There was a significant positive indirect effect for internet/social media intensity, *B* = 0.19, 95% confidence interval (CI) (0.15, 0.24); whereas the direct effect of internet/social media intensity on the intention of taking nonpharmaceutical preventive measures was significant, *B* = 0.26, 95% *CI* (0.20, 0.31), and the total effect was significant, *B* = 45, 95% *CI* (0.39, 0.51). Internet/social media intensity increased with the perceived threat of COVID-19, *B* = 0.40, 95% *CI* (0.34, 0.46), *SE* = 0.03, *p* < 0.001, which predicted more intentions of taking nonpharmaceutical preventive measures, *B* = 0.48, 95% *CI* (0.43, 0.54), *SE* = 0.16, *p* < 0.001 (see Figure 1). This test verified that the perceived threat of COVID-19 mediated the relationship between internet/social media intensity and willingness to take nonpharmaceutical preventive measures.

### 3.3. Cross-Lagged Analyses Using Both Wave 1 and Wave 2 Data

The longitudinal data were analyzed using cross-lagged panel models to determine the prospective influence of internet/social media intensity on the perceived threat of COVID-19, and the adoption of nonpharmaceutical prevention measures.

The model showed an acceptable fit, *χ*^2^ (26) = 209.22, CFI = 0.99, RMSEA = 0.01, SRMR = 0.02. Consistent with our hypothesis, internet/social media intensity predicted the perceived threat of COVID-19 (*β_T_*_1-*T*2_ = 0.19, *p* = 0.003). However, the perceived threat of COVID-19 did not predict internet/social media intensity (*β_T_*_1-*T*2_ = −0.01, *p* = 0.84). The cross-lagged effects from internet/social media intensity to the perceived threat of COVID-19 were significantly stronger than that of the opposite direction, with the unstandardized estimate = 0.19, *p* < 0.05 (see Figure 2).

In a similar vein, the model also showed an acceptable fit, *χ*^2^ (26) = 213.52, CFI = 0.98, RMSEA = 0.06, SRMR = 0.02. Internet/social media intensity predicted the adoption of nonpharmaceutical pandemic prevention measures (*β_T_*_1-*T*2_ = 0.13, *p* = 0.056). However, the adoption of nonpharmaceutical pandemic-prevention measures did not predict internet/social media intensity (*β_T_*_1-*T*2_ = −0.02, *p* = 0.78). The cross-lagged effects from internet/social media intensity to the adoption of nonpharmaceutical pandemic prevention measures were significantly stronger than that of the opposite direction, with the unstandardized estimate = 0.13, *p* = 0.056 (see Figure 3).

### 3.4. Mediation Analysis for Pharmaceutical Measures by Wave 2 Data

The mediation analyses showed that the association between internet/social media intensity and willingness to take pharmaceutical preventive measures was significantly mediated by the perceived threat of COVID-19, with an indirect effect *B* = 0.08, 95% *CI* (0.02, 0.14); whereas the direct effect of internet/social media intensity on willingness to take pharmaceutical preventive measures was significant, *B* = 0.43, 95% *CI* (0.20, 0.65), and the total effect was also significant, *B* = 0.51, 95% *CI* (0.23, 0.73). Internet/social media intensity increased with the perceived threat of COVID-19, *B* = 0.18, 95% *CI* (0.04, 0.33), *SE* = 0.07, *p* < 0.05, which predicted more willingness to take pharmaceutical preventive measures, *B* = 0.42, 95% *CI* (0.21, 0.62), *SE* = 0.11, *p* < 0.001 (see Figure 4). By measuring pharmaceutical preventive measures, this finding conceptually verified the results pattern of Wave 1.

## 4. Discussion

The current research investigated the effect of internet and social media use on preventive measures against COVID-19. By adopting a two-wave longitudinal study and recruiting a demographically representative sample in mainland China with respect to age and gender, it was verified that the intensity of internet/social media use could predict both nonpharmaceutical preventive measures (Wave 1 data) and pharmaceutical preventive measures (Wave 2 data), in which the perception of the pandemic threat served as a mediator. The cross-lagged analysis of the two waves’ data further confirmed the direction of the influence: more internet/social media use predicted more perceived threat of COVID-19, as well as more adoptions of preventive measures (nonpharmaceutical), rather than the other way around. As a response to the research question, it can be concluded that internet and social media use allowed people to make a correct judgment about the severity of pandemic and take effective preventive measures against it accordingly in the context of the People’s Republic of China.

The current study verified that the perception of the pandemic threat mediated the link between internet/social media use and taking appropriate preventive measures against the pandemic. This mediation effect showed that perception of the pandemic as a legitimate threat was the key for people to be willing to take preventive measures, even though this perception might generate negative emotions (such as fear and anxiety [41] of being infected). A previous study showed that fear of COVID-19 predicted positive behaviors change (e.g., social distancing, improved hand hygiene) [42]. However, the influence of perceptions of threat and emotions could have variable effects on health behaviors. For example, risk perception of breast cancer was found to be positively related to cancer screening [43], while cancer-related fear led to reduced cancer screening [44]. Another possibility is that negative emotions caused by the pandemic may increase preventive measures, but at the same time trigger other problems. For example, it was theorized that cyberchondria (excessive or repeated seeking of online health information) is associated with increasing levels of health anxiety or distress, creating a negative cycle of information search and a negative affect [45]. However, it was found that cyberchondria was not only associated with negative emotions (anxiety), but also related to safety behaviors (information seeking, avoidance, and hygiene) [46]. The current study did not measure the emotion aroused or accompanied by threat perception. Future research could examine more closely the different roles of threat perception and negative emotions in pandemic coping.

Besides threat perception and emotion, future research could also explore other potential factors relevant to a broader theoretical context. As the extended parallel process model (EPPM) [47] posits, the response to a threat situation is determined by four individual level factors: severity, the perception the individual has of the magnitude of the threat (similar to threat perception of the current study); susceptibility, the perception that the threat is likely to impact them; self-efficacy, the perception the individual is competent to handle the threat; and response efficacy, the perception that the action, if carried out, will successfully control the threat. The EPPM predicts that danger control (i.e., taking nonpharmaceutical and pharmaceutical preventive measures in the current study) takes place when individuals perceive both high severity and susceptibility, and they feel capable of taking effective actions. If individuals do not feel competent to take effective measures, they may turn to controlling their negative emotions, even if severity and susceptibility remain high. Returning to the context of the pandemic, this model implies that if people think the pandemic has gone beyond their competence to deal with, or they cannot find effective measures against it, they may turn to controlling their negative emotions, such as fear and anxiety, rather than taking appropriate preventative actions. Future studies could further explore these boundary conditions by considering complex factors involving emotions, probability, and actions.

The current study may shed light on the social effects of fact-checking in mainland China, which need to be considered when generalizing the findings of the current study. The credibility of information about the pandemic is the premise of findings of the study. To combat misinformation on social media, many online platforms in mainland China, such as WeChat and Weibo, have a customer report feature as the main strategy of fact-checking services, and continue to use and consult with information from trusted partners, including the government, academic institutions, and NGOs, when reviewing content [48,49]. Transparency and proactive communication are essential to early detection and containment of pandemics [50], which led to criticism of the Chinese government’s delayed early warning of the pandemic before its outbreak [51]. However, it also must be acknowledged that China’s information-governance policies demonstrated effectiveness and immediacy in containing COVID-19 soon after the pandemic outbreak [52]. China’s information governance, using strategies such as fact-checking by authorities, can be viewed like the movements of an “elephant”. Compared with societies without fact-checking, China’s top-down media environment responds slowly, due to ignoring or suppressing different voices on one hand. However, on the other hand, it can rapidly reduce misinformation and effectively disseminate the right message once this “elephant” wakes up to a sound judgment of the situation. Fact-checking does not work alone, but rather interacts with ideology, and public expectation of the correct role of the government, authoritative organizations, and public figures [53]. The ability and motivation of these authorities to consistently provide a truthful view across a variety of situations can be open to questioning. Does this mean that the findings of the current study could only be applied to China? Considering that Twitter also adopted a fact-checking policy [54], we expect that the current findings may also be generalized to other parts of the world. This is an open question, worth exploring in future studies.

Another related issue is that there may be massive variability in how much a given individual is willing and able to source credible information on the internet. Because of information biases such as confirmation bias [55] and the filter bubble effect [56], greater internet/social media use does not provide assurance that an individual will obtain more credible information. If people cannot take full advantage of the internet and social media, they may generate false judgments of the situation (e.g., underestimate the infectivity of or overestimate one’s immunity to the pandemic), and take incorrect actions (e.g., refuse to wear masks) as a result. Once incorrect judgments and actions grow by socializing with like-minded people on the internet and social media, they may facilitate social extremism and polarization [57] through the “echo chamber effect” [58], which is an accentuation of differences as a function of algorithms on social media that expose people to content based on previous preferences [59], and then transforms into harmful collective behaviors (e.g., the coronavirus challenge of licking public toilet seats on TikTok [60]). Future studies could examine the issue further by exploring the effect of accessing specific categories or sites of credible versus noncredible information. For instance, regarding the severity of the pandemic, the authenticity of information may moderate the relationship between internet/social media intensity and the perceived threat of COVID-19 found in the current study. In addition, regarding preventive measures against the pandemic, the authenticity of information may moderate the relationship between the perceived threat of COVID-19 and intentions of taking appropriate measures.

It is also beneficial to keep in mind that the regulatory systems to check people’s compliance with preventive measures were culturally affected. For example, a recent cross-cultural study analyzed the antipandemic policy strategies of eight European countries, and showed that whether a given country deployed strong rule-based regulatory systems was associated with power distance and uncertainty avoidance at the national level [61]. Likewise, future research could examine the generality of the current study by exploring how the current findings are replicated in other societies with different regulatory schemes of information governance online, values [62], social mentality [63], and ideology [64]. East Asian societies in 2020 were particularly effective in pandemic control for a variety of reasons, including culture and previous experiences with MERS, rather than simple compliance with top-down authoritarianism or state-controlled fact-checking [65].

## 5. Conclusions

Our results indicated that, in the context of mainland China, internet/social media use was positively associated with nonpharmaceutical and pharmaceutical antipandemic preventive measures, and these links were mediated by an increased level of perceived threat. The current findings contribute to a better understanding of the positive role of the internet and social media in China’s combatting COVID-19 pandemic, in which fact-checking is effectively enforced by both governmental and nongovernmental actors to ensure a healthy media environment during the COVID-19 pandemic.

## Figures and Tables

**Figure 1 healthcare-10-00113-f001:**
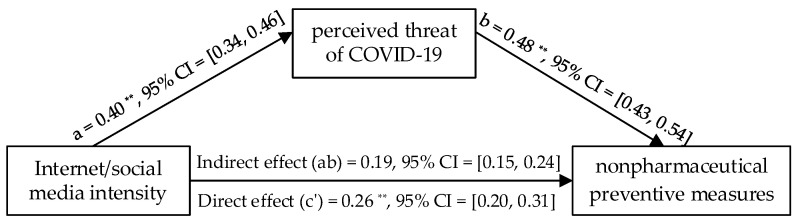
Results of the mediation model. Note: *n* = 1006. ** *p* < 0.01.

**Figure 2 healthcare-10-00113-f002:**
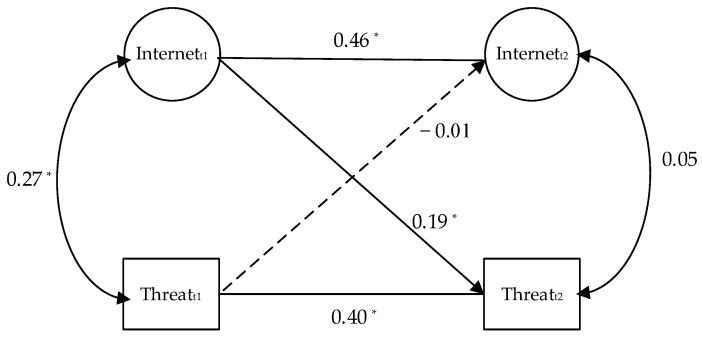
Result of the cross-lagged analyses of the relationship between internet/social media intensity and the perceived threat of COVID-19. Note. Internet = internet/social media intensity; Threat = the perceived threat of COVID-19; t1 = wave 1, t2 = wave 2. * *p* < 0.05.

**Figure 3 healthcare-10-00113-f003:**
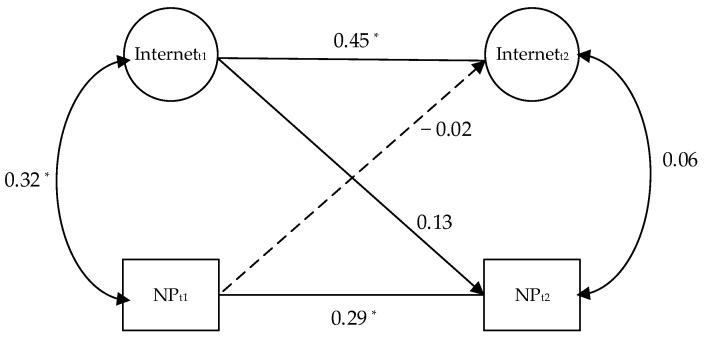
Result of the cross-lagged analyses of the relationship between internet/social media intensity and nonpharmaceutical preventive measures. Note: Internet = internet/social media intensity; NP = nonpharmaceutical preventive measures; t1 = wave 1, t2 = wave 2. * *p* < 0.05.

**Figure 4 healthcare-10-00113-f004:**
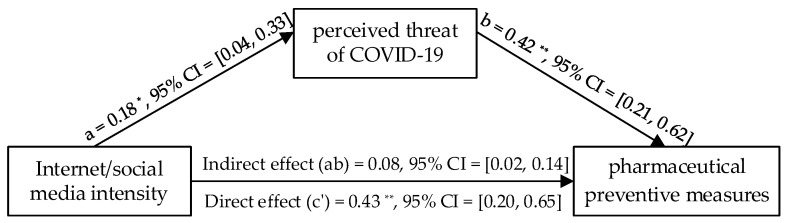
Results of the mediation model. Note: *n* = 218. * *p* < 0.05, ** *p* < 0.01.

**Table 1 healthcare-10-00113-t001:** Means, standard deviations, and intercorrelations among variables (Wave 1).

Variable	1	2	3	4	5	6	7	8	*M*	*SD*
1. Internet (*n* = 1016)	-								5.11	0.92
2. Threat (*n* = 1017)	0.35 **	-							6.30	0.91
3. OT (*n* = 1014)	0.03	−0.05	-						0.48	0.50
4. NPM (*n* = 1016)	0.41 **	0.56 **	−0.04	-					6.25	0.94
5. Age (*n* = 1014)	−0.18 **	0.08 **	−0.17 **	0.00	-				37.25	10.32
6. Gender (*n* = 1015)	0.04	−0.03	0.14 **	0.01	−0.28 **	-			0.52	0.50
7. Education (*n* = 1014)	0.19 **	−0.02	0.11 **	0.02	−0.25 **	0.10	-		4.80	0.71
8. Status (*n* = 1015)	0.20 **	−0.05	0.01	0.01	−0.07 *	0.03	0.20	-	5.42	1.55

Note: Internet = internet/social media intensity; Threat = the perceived threat of COVID-19; OT = online time; NPM = nonpharmaceutical preventive measures. OT: 0 = numbers of hours per day, 1 = almost all waking hours. Gender: 0 = male, 1 = female. * *p* < 0.05, ** *p* < 0.01.

**Table 2 healthcare-10-00113-t002:** Means, standard deviations, and intercorrelations among variables (Wave 2).

Variable	1	2	3	4	5	6	7	8	*M*	*SD*
1. Internet (*n* = 220)	—								4.94	0.84
2. Threat (*n* = 220)	0.13	—							6.35	0.84
3. OT (*n* = 220)	0.04	0.00	—						0.53	0.50
4. PM (*n* = 220)	0.32 **	0.25 **	−0.05	—					5.27	1.39
5. Age (*n* = 220)	−0.19 **	0.18 **	−0.00 **	−0.12	—				36.40	8.98
6. Gender (*n* = 219)	0.07	−0.04	0.04	−0.06	−0.28 **	—			0.51	0.50
7. Education (*n* = 219)	0.18 **	−0.10	0.11 **	0.17 *	−0.21 **	0.02	—		4.83	0.61
8. Status (*n* = 220)	0.39 **	−0.01	−0.03	0.12	−0.27 *	0.11	0.26 **	—	5.63	1.51

Note: Internet = internet/social media intensity; Threat = the perceived threat of COVID-19; OT = online time; PM = pharmaceutical preventive measures. OT: 0 = numbers of hours per day, 1 = almost all waking hours. Gender: 0 = male, 1 = female. * *p* < 0.05, ** *p* < 0.01.

## Data Availability

The data presented in this study are available upon request from the corresponding author.

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
