# Peer review of "The Active Role of the Internet and Social Media Use in Nonpharmaceutical and Pharmaceutical Preventive Measures against COVID-19"

_healthcare, 2022, doi:10.3390/healthcare10010113_

Round 1
Reviewer 1 Report
This study examined the relationship between internet usage, perceived threat, and the use of preventive measures. It addressed an important and timely research question. Given the data were collected in China, the authors should discuss if the findings will hold in other countries.
Author Response
Dear reviewer,
Thank you for your useful suggestions. We have responded to your comments point by point in detail. You could refer to the uploaded file.

Reviewer 2 Report
work deals with a very interesting and debated problem linked to the spread of fake news in covid-19 web through the internet and social media.
In general, overall, the work shows two main limitations. The first is of a temporal nature, as it refers to the central part of 2020, a period in which scientific knowledge on Covid-19 was evolving. The second is related to the sample. In the second phase, only 220 of the 797 subjects who responded in the first phase responded. This removes statistical concern about the results and does not allow the statements made by the authors at the end of the conclusions to be considered as proven.
In particular the introduction is extremely long; it is essential that many parts are eliminated (from row 33 to 38; from 67 to 73; from 87 to 91; from 100 to 139).
Author Response

(The authors gave the same response as above.)

Reviewer 3 Report
My review of “The Active Role of the Internet and Social Media Use in Non-pharmaceutical and Pharmaceutical Preventive Measures against COVID-19”
The authors investigated the relationship between Internet/Social media use and non-pharmaceutical preventive measure also considering perceived threat as a mediator. The paper should be improved in my opinion based on the following comments.
I am perfectly aware that remaining updated about covid-19 literature is not an easy task currently, but I do think that strengthening the literature background would be essential to avoid publishing an "outdated" paper.
For this reason, I suggest the authors mention the following papers:
- https://doi.org/10.1108/OIR-09-2020-0417 (about Covid-19 infodemic)
- https://doi.org/10.1371/journal.pone.0243704; https://doi.org/10.1002/hbe2.233 (two good papers related to Covid-19 cyberchondria)
- https://doi.org/10.3390/covid1010020 (the paper focused on compliance to preventive measures - like the ones considered in your paper - and highlighted the role of risk perception, personality, and well-being)
- https://doi.org/10.3390/covid1010015 (the paper stressed the relationship between knowledge of individuals and COVID-19 safety practices)
- https://doi.org/10.3390/covid1010005 (it would be good to mention that the regulatory systems to check people compliance in preventive measures were culturally affected)
- https://doi.org/10.3390/fi13110286 (among non-pharmaceutical preventive measures, contact tracing systems adoption should be at least mentioned)
- Lines 91-94 could be better supported by referencing good gossip, pro-social gossip dynamics, reciprocity norm, and reputation-related gains: https://doi.org/10.1037/a0026650;
"...higher perceived threat of the pandemic, which would then lead to effective preventive measures against the pandemic". Well, it is not that simple. Although I suggested you cite a paper that highlights the role of risk perception (among other factors) in compliance with preventive measures, the authors should also refer to the extended parallel process model (https://doi.org/10.1080/03637759209376276). To put it simply, a perception of threat that is too high compared to the individuals' coping capacity can lead to dynamics of avoidance or resolve one's cognitive dissonance in a different way from compliance.
Line 136. "our hypothesized causal relationship". The design of your study does not allow for causal explanations. I would suggest smoothing this claim.
Line 145. How exactly did the commercial survey company reach the participants? Which channels were used?
Were the items you rely on to measure Internet/social media intensity used in other studies that support the validity of this way of proceeding?
As for the perceived threat of COVID-19, I have some concerns. Why did the authors not use an already validated risk perception scale like those of Wilson (https://doi.org/10.1111/risa.13207) or Prati (https://doi.org/10.1111/j.1539-6924.2010.01529.x)? These two scales were also adapted to account for Covid-19. Please justify your choice.
Please include a statement regarding the assumptions for parametric analysis (e.g., normality).
The model presented in Figure 1 needs further information about how much variance is explained by the model. Moreover, the indirect standardized effect should be presented as well. The same should be done for the model in Figure 4.
Discussion: In general, the paper seemed to underestimate the role of information search biases. Higher use of Internet/Social media use may exacerbate them. For instance, confirmation bias and filter bubble effect are not necessarily decreased by higher use of the Internet. Thus, as a future perspective, the authors could try to include more of these aspects in the discussion.
Author Response

(The authors gave the same response as above.)

Round 2
Reviewer 2 Report
The problem relating to the significant difference between the two sample is not solved. The literature reported to support the validity of the comparison between these samples il very old (the most recent is from 1990s). The doubts about the metodological correctness of the study and significance of the results remain. Some observations in the reply are only obvious (on the internet you can find also correct information....).
Author Response
Thanks for your concerns, comments and suggestions. It seems that you have two points. The first one is about the participants attrition, while the second one is about the significance of our study. We respond to the two points as in the attached file.

Reviewer 3 Report
The authors were able to convincingly answer all my points and now the manuscript appears substantially improved.
Author Response
Comment: The authors were able to convincingly answer all my points and now the manuscript appears substantially improved.
Answer: Many thanks! The maunscript has been imporved a lot due to your detailed suggestions.